# Determination of Pesticide Residues in Fresh Fruits in the Serbian Market by LC-MS/MS

**DOI:** 10.3390/foods14101828

**Published:** 2025-05-21

**Authors:** Isidora Kecojević, Danica Mrkajić, Vladimir Tomović, Biljana Bajić, Milana Lazović, Ana Joksimović, Mila Tomović, Aleksandra Martinović, Dragan Vujadinović, Srđan Stefanović, Vesna Đorđević

**Affiliations:** 1Faculty of Technology Novi Sad, University of Novi Sad, 21000 Novi Sad, Serbia; isidora.kecojevic@abiotechlab.rs (I.K.); danica.mrkajic@abiotechlab.rs (D.M.); milana.lazovic@abiotechlab.rs (M.L.); 2A Bio Tech Lab d.o.o., 21208 Sremska Kamenica, Serbia; biljana.bajic@abiotechlab.rs (B.B.); ana.joksimovic@abiotechlab.rs (A.J.); 3Technical School “Pavle Savić”, 21000 Novi Sad, Serbia; mila.tomovic4@gmail.com; 4Faculty for Food Technology, Food Safety and Ecology, University of Donja Gorica, Donja Gorica, 81000 Podgorica, Montenegro; aleksandra.martinovic@udg.edu.me; 5Faculty of Technology Zvornik, University of East Sarajevo, 75400 Zvornik, Bosnia and Herzegovina; dragan.vujadinovic@tfzv.ues.rs.ba; 6Institute of Meat Hygiene and Technology, 11040 Belgrade, Serbia; srdjan.stefanovic@inmes.rs (S.S.); vesna.djordjevic@inmes.rs (V.Đ.)

**Keywords:** fruits, pesticide residues, maximum residue levels, food safety

## Abstract

The concentrations of pesticide residues were determined in 2164 samples of 46 fruit species, collected over a 4-year period. Fruits originated from 59 countries, including Serbia (N = 199). Pesticide residues were determined by liquid chromatography–tandem mass spectrometry (LC-MS/MS) after extraction using a modified QuEChERS protocol. A total of 173 pesticide residues were detected. Of the fruit samples, 62.57% had pesticide residues at or above 0.01 mg/kg, and 4.67% exceeded the maximum residue limits (MRLs) set by the Serbian regulation. MRL values were most often exceeded in pomegranate and citrus fruits (grapefruit and mandarin). The most frequently found pesticide was imazalil (detected in 624 samples, 28.84%), with the highest concentration (93.870 mg/kg) found in a grapefruit sample. Multiple pesticides were detected in 50.92% of the fruit samples, and two grapefruit samples contained up to 44 pesticide residues.

## 1. Introduction

Pesticides play a significant role in food production. They help to protect crops and grow more yields; their use can also increase the number of times each year a crop can be grown on the same land. Pesticides are used to protect crops against insects, weeds, fungi, and other pests. Since they are designed to be biologically active, pesticides are potentially toxic to humans and can have both acute and chronic health effects, depending on the quantity and ways in which a person is exposed [1,2,3,4,5]. People who are both directly and indirectly exposed to pesticides may suffer acute toxic effects, including suicide attempts, mass poisonings from contaminated food, chemical accidents in the industry, and occupational exposure in the agricultural industry, as well as a number of serious chronic diseases, including cancer, asthma, diabetes, Parkinson’s disease, leukemia, and cognitive impairment. For example, pesticide use causes three million poisonings, 220,000 deaths, and about 750,000 chronic illnesses every year worldwide, most of them occurring in developing countries [6,7]. For this reason, pesticide residue levels in food are regulated by national and European legislation. These comprehensive legislative frameworks define rules for the approval of active substances, their uses in plant protection products, and their permissible residues in food. To ensure a high level of consumer protection, legal limits, or so-called ‘maximum residue levels’ (MRLs), are established in Regulation (EC) No 396/2005 [8]. European Union harmonized MRLs are set for more than 1300 pesticides, covering 378 food products/food groups [9]. This regulation is being continuously amended by several Regulations and Commission Regulations to update the commodities included and their residue levels, based on the most recent knowledge from the Member States, the European Food Safety Authority (EFSA), and the European Commission. In Serbia, the regulation on the “maximum residue levels of pesticides in food” has changed a few times over the past 10 years [10,11,12], with the aim of aligning national legislation with EU regulation. The latest Serbian Regulation [13] on the “maximum residue levels of pesticides in food” is fully harmonized with the European Union Regulations.

There have been many surveys of pesticide residues in fruits/food recently [14,15,16,17,18,19,20,21,22,23,24,25,26,27,28,29,30,31,32,33,34,35,36,37,38,39,40,41,42,43,44,45,46,47,48,49,50,51,52,53,54,55,56,57,58,59,60,61,62,63,64,65,66,67,68,69,70,71,72,73]. A summary of the studies published in the available literature dealing with pesticide residues in fruit (from 2010 onwards) is presented in Appendix A. The overall conclusion of these studies is that there is a widespread pesticide presence currently in fruits/food in general. The pesticides are frequently detected above the limit of quantification (LOQ); however, MRL is seldom exceeded. Some pesticide residues were detected at concentrations above their MRLs. In addition, pesticides that are found in these studies are banned and/or unauthorized in many countries because of their high toxicity. Therefore, pesticide residue control is an important activity intended to prevent, reduce, or eliminate chemical hazards in food.

Pesticide residue determination using analytical methods involves the main steps such as sample preparation (homogenization, extraction, and clean-up), separation, detection, and data analysis [5]. The extraction procedure is a crucial part of analytical methods and has a great influence on the correct quantification of pesticide residues. The most common extraction methods are liquid–liquid extraction (LLE), solid-phase extraction (SPE), matrix solid-phase dispersion (MSPD), quick, easy, cheap, effective, rugged, and safe (QuEChERS) extraction, and solid-phase microextraction (SPME) [5]. The QuEChERS sample preparation technique is extensively used for the multiclass or multiresidues analysis of different types of pesticides mainly in agriculture/foodstuff/fruit/vegetable [5,14,15,16,17,18,19,20,21,22,23,24,25,26,27,28,29,30,31,32,33,34,35,36,37,38,39,40,41,42,43,44,74]. Numerous detection methods have been developed over the past two decades for the determination of pesticide residues in food samples. The conventional analytical methods are gas chromatography (GC) and high-performance liquid chromatography (HPLC) coupled with various detectors [electron capture detector (GC-ECD), flame photometric detector (GC-FPD), nitrogen phosphorus detector (GC-NPD), and flame ionization detector (GC-FID); UV detection (HPLC-UV), fluorescence detector (HPLC-FD), and diode-array detector (HPLC-DAD)]. Moreover, liquid chromatography–mass spectrometry (LC-MS) is a widely utilized method for multi-pesticide residue analysis. MS (and tandem mass spectrometry—MS/MS) detection system presents high sensitivity and selectivity without derivatization. Identification and confirmation can be conducted in a single step when applied to LC [5,14,15,16,17,19,20,21,22,23,25,26,27,29,30,32,33,35,36,37,41,42,43,44,45,46,47,48,49,50,51,52,53,74].

The objective of this study was to investigate the concentrations of pesticide residues in fruits, which are collected as a part of the national monitoring programme for pesticide residues in Serbia, and to compare these levels with maximum residue levels established by the Serbian Regulation [10,11,12].

## 2. Materials and Methods

Over a 4-year period (2016–2019), concentrations of pesticide residues were determined in 2164 samples of fresh fruits. The analyses were conducted by an accredited state laboratory (A Bio Tech Lab).

Primary samples of each fruit were taken from eight different packages, and six fruits were taken randomly from each package. The following parts of the fruit were used for homogenization: whole product with the stem (currants); whole product after removal of the stem (apple, apricot, avocado, banana, carambola, cashew, clementine, date palm, fig, grapefruit, Japanese apple, kiwi, kumquat, lemon, lime, lychee, mandarin, mango, orange, passion fruit, peach, peach (nectarine), pear, pitaya, plum, pomegranate, pomelo, quince, sour cherry, sweet cherry); whole product with the shell (chestnut); whole product after removal of the shell (almond, Brazil nut, coconut, hazelnut, peanut, pistachio, walnut); whole product after removal of the crown (pineapple); whole product after removal of the caps, crown, and stems (aronia, blackberry, blueberry, grape, raspberry, strawberry) [8,10,11,12,13].

The method for sample preparation and analysis of the concentrations of pesticide residues in the collected fruits was conducted as described in detail in Kecojević et al. [74]. This method uses a modified QuEChERS (Quick, Easy, Cheap, Effective, Rugged, Safe) protocol without a clean-up step, based on the sample dilution approach, followed by liquid chromatography coupled with the tandem mass spectrometry (LC-MS/MS). Control spike samples were prepared using the following procedure. After grinding and homogenization, 10 g fruit samples were measured into 50 mL polypropylene centrifuge tubes (PP tubes). Then, 100 μL of internal standard and an appropriate solution of the stock pesticide standard were added so that the final concentrations were 5 and 50 μg/kg for the fruit samples. A total of 10 mL of extraction solvent (ACN) was added to the samples. After homogenization by hand shaking (1 min), QuEChERS Mix I (a mixture of 4 g magnesium sulphate, 1 g sodium chloride, 0.5 g disodium hydrogen citratesesquihydrate, and 1 g sodium citrate dihydrate) was added. The whole content of the tube was hand shaken for another 1 min and centrifuged for 5 min at 3000 rpm. Then, 500 μL of the solution was transferred into a vial, and 500 μL MP-A and 10 μL of 5% formic acid in ACN were added. Prior to the analysis, all extracts were filtrated through 0.45 μm polytetrafluoroethylene (PTFE) filters. Pesticides were analysed using Thermo Scientific UHPLC (ultra-high performance liquid chromatography) system, Dionex UltiMate 3000, coupled with an MS detector triple-quadrupole Thermo Scientific TSQ Quantum Access Max (Thermo Fisher Scientific Inc., Waltham, MA, USA) with electrospray ionization. The mass spectrometer was operated in positive mode with the following parameters: spray voltage 5.0 kV, vaporizer temperature 280 °C, ion transfer tube temperature 300 °C, auxiliary gas (nitrogen) pressure 15 arb. units, and sheath gas pressure 55 arb. units. The collision gas was argon (pressure set at 1.5 mTor). A selected reaction monitoring (SRM-EZ) method was applied. Chromatographic separation was carried out with reverse-phase Thermo Scientific Hypersil Gold aQ, (100 mm 2.1 mm, i.d. 1.9 μm) analytical column, under following gradient profile: 0–2.1 min, 95% MP-A; 2.1–25.0 min, 95–5% MP-A; 25.0–35.0 min, 5% MP-A; 35.0–35.1 min, 5–95% MP A; 35.1–40.8 min, 95% MP-A. The flow rate of the mobile phase was 0.3 mL/min, the injection volume was 6 μL, and the column oven temperature was 40 °C. All samples were filtrated through 0.45 μm PTFE membrane filters before chromatographic analysis.

The results were evaluated according to the reporting limit (RL) and MRLs established by Serbian regulation. The LOQ for all pesticide residues was 0.005 mg/kg (LOQ for cabbage, Kecojević et al. [74]), while the reporting limit (RL) was 0.01 mg/kg. Generally, Serbian, as well as EU, MRLs for pesticide residues in fruits are in the range of 0.01–10 mg/kg, depending on the compound. Only for a few pesticides are MRLs up to 15 and 20 mg/kg.

## 3. Results and Discussion

Detailed characteristics like common name, country of origin, and number of samples (without and with pesticide residues) of the analysed samples are shown in Table 1. All pesticide residues at or above the reporting limit (RL ≥ 0.01 mg/kg) are presented.

In this study, a total of 2164 samples of fresh fruits were analysed for pesticide residue. The number and percentage of samples collected per year are shown in Figure 1. The origin of the samples is shown in Figure 2. Evaluation of the obtained results for the 2164 different fruit samples (Table 1) showed that 62.57% (1354 out of 2164 samples) contained pesticide residues (RL ≥ 0.01 mg/kg), while 37.43% (810 out of 2164 samples) contained no pesticide residues (RL < 0.01 mg/kg). All samples of aronia (N = 1), Brazil nut (N = 2), carambola (N = 2), cashew (N = 1), chestnut (N = 2), fig (N = 1), lychee (N = 1), passion fruit (N = 1), pistachio (N = 1), pitaya (N = 2), and quince (N = 1) were pesticide-free. The detection rates (when the sample size “N” is greater than 30) of pesticide residues in peach (nectarine) (N = 36), pineapple (N = 41), strawberry (N = 46), kiwi (N = 50), peach (N = 56), pomegranate (N = 66), pear (N = 95), grapefruit (N = 115), grape (N = 152), banana (N = 168), mandarin (N = 194) lemon (N = 217), orange (N = 324), and apple (N = 351) samples were 36.11%, 73.17%, 76.09%, 34.00%, 42.86%, 63.64%, 71.58%, 88.70%, 34.87%, 58.33%, 69.07%, 80.65%, 69.75%, and 64.10%, respectively. All pesticide residues detected in almond (N = 6), apricot (N = 15), aronia (N = 1), banana (N = 168), blackberry (N = 2), blueberry (N = 5), Brazil nut (N = 2), carambola (N = 2), cashew (N = 1), chestnut (N = 2), clementine (N = 16), coconut (N = 1), currants (N = 2), date palm (N = 8), fig (N = 1), hazelnut (N = 5), Japanese apple (N = 9), kiwi (N = 50), kumquat (N = 4), lychee (N = 1), mango (N = 24), passion fruit (N = 1), peach (N = 56), peach (nectarine) (N = 36), peanut (N = 8), pistachio (N = 1), pitaya (N = 2), plum (N = 26), pomelo (N = 12), quince (N = 1), raspberry (N = 8), strawberry (N = 46), sweet cherry (N = 20), and walnut (N = 6) were below or at the MRLs. The MRLs for pesticide residues were exceeded in 101 out of the 2164 (4.67%) samples: apple (12 out of the 351 samples, 3.42%; Bosnia and Herzegovina: N = 2, North Macedonia: N = 2, Poland: N = 3, Serbia: N = 5), avocado (1 out of the 19 samples, 5.26%; Peru: N = 1), grapefruit (21 out of the 115 samples, 18.26%; South Africa: N = 4; Turkey: N = 17), grapes (2 out of the 152 samples, 1.32%; North Macedonia: N = 2), lemon (4 out of the 217 samples, 1.84%; Argentina: N = 1, Turkey: N = 2), lime (4 out of the 28 samples, 14.29%; Guatemala: N = 1, Mexico: N = 2, The Netherlands: N = 1), mandarin (14 out of the 194 samples, 7.22%; Spain: N = 2, Swaziland: N = 1, Turkey: N = 11), orange (7 out of the 324 samples, 2.16%; Spain: N = 1, Turkey: N = 6), pear (6 out of the 95 samples, 6.32%; Poland: N = 2, Serbia: N = 4), pineapple (3 out of the 41 samples, 7.32%; Colombia: N = 1, Costa Rica: N = 2), pomegranate (26 out of the 66 samples, 39.39%; Turkey: N = 26), and sour cherry (1 out of the 14 samples, 7.14%; Serbia: N = 1).

The frequency of the detected pesticide residues in 2164 fruit samples is shown in Table 2. A total of 173 pesticide residues (distributed as follows: 49.13% insecticides, 35.84% fungicides, and 15.03% herbicides) were detected in all the fruit samples. Imazalil, thiabendazole, pyrimethanil, imidacloprid, carbendazim, acetamiprid, boscalid, fludioxonil, chlorpyrifos, prochloraz, pyriproxyfen, tebuconazole, propiconazole, prothioconazole, azoxystrobrin, methoxyfenozide, and pyraclostrobin were the pesticide residues most frequently found (occurrence in more than 90 analysed samples, >4%) and were detected in 624 (28.84%), 337 (15.57%), 245 (11.32%), 213 (9.84%), 185 (8.55%), 171 (7.90%), 154 (7.12%), 154 (7.12%), 145 (6.70%), 143 (6.61%), 138 (6.38%), 116 (5.36%), 113 (5.22%), 97 (4.48%), 96 (4.44%), 94 (4.34%), and 93 (4.30%) samples, respectively. Of the 173 pesticide residues, 64 (36.99%) of them were detected at least once in fruit samples at levels higher than MRLs. A total of 309 pesticide residues (133 in pomegranate, 43.04%; 95 in grapefruit, 30.74%; 33 in mandarin, 10.68%; 13 in apples, 4.21%; 11 in orange, 3.56%; 7 in pear, 2.27%; 5 in lime, 1.62%; 4 in lemon, 1.29%; 3 in pineapple, 0.97%; 2 in grape, 0.65%; 2 in sour cherry, 0.65; 1 in avocado, 0.32%) were found in the 101 fruit samples containing residues above MRLs. The other 109 (63.01%) pesticide residues did not exceed their MRL values. The most frequent pesticide residues found to exceed the MRL were butoxycarboxim (100%, 5 out of 5 samples), carbofuran (100%, 1 out of 1 sample), deltamethrin (60.00%, 12 out of 20 samples), fenvalerate (60.78%, 31 out of 51 samples), formothion (58.82%, 10 out of 17 samples), iprodione (66.67%, 2 out of 3 samples), mepanipyrim (61.54%, 8 out of 13 samples), mepronil (100%, 1 out of 1 sample), nuariomol (66.67%, 2 out of 3 samples), oxamyl (100%, 10 out of 10 samples), piperonyl-butoxide (50.00%, 1 out of 2 samples), pirimiphos-methyl (83.33%, 10 out of 12 samples), promecarb (71.43%, 5 out of 7 samples), prometryn (75.00%, 3 out of 4 samples), sulfentrazone (50.00%, 2 out of 4 samples), tebuthiuron (63.64%, 7 out of 11 samples), terbutryn (50.00%, 3 out of 6 samples), and tricyclazole (100%, 3 out of 3 samples). Among the most detected pesticide residues, imazalil was found at the highest concentration (93.349 mg/kg, 18.67 times higher than MRL) in grapefruit.

Pesticide residues in fruit samples were reported in many countries. The following presents the available results on pesticide residues in fruits where the levels exceeded the MRLs and where the number of investigated samples was equal to or higher than 50, using a multi-residue analytical method. Abdallah et al. [14] performed 42 pesticide residue analyses (10 of which were detected) on 200 date samples collected from different large markets in the Al-Qassim region in Saudi Arabia and reported that 15 samples (7.5%) exceeded the MRLs. The acetamiprid, carbendazim, carbofuran, difenoconazole, indoxacarb, malathion, and oxadiazon residues exceeded the MRLs. The results revealed that the carbofuran, imidacloprid, acetamiprid, and difenoconazole residues were the most frequently detected pesticides. In Lebanon, Abou Zeid et al. [15] analysed 76 pesticide residues (19 of which were detected) in 128 loqaut samples gathered from markets between 2017 and 2019. The authors reported that pesticide concentration exceeded the MRLs in 39.84% of the samples. In a Turkish study [16] on lemon samples (N = 100), including 355 pesticide residues (16 of which were detected), 29% of the samples exceeded the MRLs. The buprofezin, chlorpyrifos-methyl, and metamitron residues exceeded the MRLs. The most frequently detected pesticides in lemon fruits were chlorpyrifos-methyl, metamitron, buprofezin, pyriproxyfen, and malathion. In another Turkish study, Bakırcı et al. [17] analysed a total of 1423 fruits and vegetables (573 samples of fruits and 850 samples of vegetables) from the Aegean region of Turkey. The samples were analysed to determine the concentrations of 186 pesticide residues (80 of which were detected). These researchers found that 8.4% of the fruit samples contained pesticide residues above the MRLs. Acetamiprid, chlorpyriphos, and carbendazim were the most detected pesticide residues. El Hawari et al. [18] examined 212 apple samples in Lebanon during 2012–2016. Apple samples were tested for up to 39 pesticide residues. They found that 61% of pesticide residues were higher than the MRLs. MRL exceedances were detected for chlorpyrifos, methidathion, and lambda-cyhalothrin. The most frequently detected pesticide residues were chlorpyrifos, methidathion, cypermethrin, lambda-cyhalothrin, myclobutanil, and diazinon. Further, El-Mageed et al. [19] examined 5381 fruit samples and 4343 vegetable samples in the United Arab Emirates (monitoring programme during 2019). Out of the 343 pesticides tested (93 of which were detected), 34 pesticides were found above the MRLs. The MRLs were exceeded in 5.43% of the fruit samples. The most commonly detected pesticides in the fruit samples that exceeded the established MRLs were bifenazate, spirodiclofen, carbendazim, and acetamiprid. Gad Alla et al. [20] carried out a survey of 177 samples of fruits collected from Egyptian markets from July to December 2010. The results indicated that 17.5% of the samples contained residues above the MRLs. Of the 215 pesticides scanned, 51 pesticides were detected. The most detected pesticide groups were pyrethroids, followed by organophosphates, benzimidazoles, neonicotinoid, carbamates, and triazoles. Organophosphates were the most violated group, followed by pyrethroids. Golge and Kabak [21] determined 172 pesticide residues (16 of which were detected) in 280 table grape samples, collected in four provinces of Turkey from August to October 2016. Residues above the MRLs were found in 20.4% of the samples. The most prevalent pesticide residues were azoxystrobin, chlorpyrifos, boscalid, and cyprodinil. Two pesticides (carbendazim and chlorpyrifos) exceeded MRLs in the tested samples. In a study by Li et al. [22], 142 pesticide residues (18 of which were detected) were detected in 245 strawberry samples gathered from markets in Beijing, China, from June 2017 to May 2018. Two pesticides (carbofuran and dimethomorph) exceeded the MRLs in the tested samples (1.63%). Carbendazim, pyrimethanil, and azoxystrobin were the most frequently detected pesticides in strawberry samples. A study by Li et al. [23] focused on pesticide contamination of citrus fruits (N = 2922) from China (monitoring programme from 2013 to 2018). MRL exceedances were recorded in 111 (3.8%) citrus fruit samples. Out of the 106 pesticides tested, 40 pesticides were found, eight of which (bifenthrin, carbofuran, cyhalothrin, difenoconazole, fenpyroximate, isocarbophos, profenofos, and triazophos) had residues that violated the MRLs. The most frequently detected pesticides were chlorpyrifos, prochloraz, carbendazim, profenofos, and acetamiprid. Likudis et al. [24] conducted a study during 2011–2012 analyzing the residues of pesticides in samples of Greek apples (N = 80). A total of 12 pesticide residues were detected out of the 51 active substances that were investigated. The highest detection rates were observed for chlorpyrifos, quinalphos, and parathion. Only 2 of the 80 investigated samples (2.5%) contained pesticide residues (parathion-methyl) in levels exceeding the MRLs. Osaili et al. [25] evaluated the pesticide residues in imported fruits (N = 4513) collected from Dubai ports (United Arab Emirates) between 2018 to 2020 during a monitoring campaign. Out of 262 pesticide reference standards that were analysed, a total of 81 pesticides were present in fruit samples. A total of 26.8% of samples contained pesticide residues above the MRLs. Chlorpyrifos, carbendazim, cypermethrin, and azoxystrobin were the most frequently detected pesticides above the MRLs. Radulović et al. [26] investigated the occurrence of pesticide residues on citrus fruits imported to Serbia (February 2022). The analysis of 76 citrus fruit samples revealed the presence of 23 different pesticides. The MRL values of pesticide residues were exceeded in 21 samples (28%). The most detected pesticides were imazalil, azoxystrobin, dimethomorph, boscalid, pyrimethanil, and thiabendazole. The MRL values of the pesticide residues were exceeded for azinphos-methyl, chlorpyrifos, dimethomorph, fluazifop, imazalil, picoxystrobin, prochloraz, and propiconazole. In Poland, Sójka et al. [27] determined 177 pesticide residues in 121 samples of fresh and frozen strawberries from the 2012 harvest season. The MRL exceedances were recorded in five (4.13%) samples. Pyraclostrobin was the most commonly detected pesticide, followed by cyprodinil, fludioxonil, and boscalid. In Mexico, Suárez-Jacobo et al. [28] diagnosed pesticide residues in oranges from Nuevo Leon (May 2014). The results showed the presence of 15 pesticide residues out of 93 analysed compounds. Of the 100 samples analysed, 11 samples exceeded the MRLs. The pesticides above the MRLs were chlorpyrifos-ethyl, dimethoate, formalathion, and methidation. The pesticide residues most frequently found were spirodiclofen, chlorpyrifos-ethyl, and malathion. Wang et al. [29] assayed pesticide residues in litchi from China during the period from May 2019 to August 2020. A total of 37 pesticide residues (23 of which were detected) were monitored in 268 litchi samples. Out of the 268 samples collected, 53 samples (19.8%) contained pesticide concentrations that exceeded the MRLs. Pesticides with the highest exceedance rates included pyraclostrobin, lambda-cyhalothrin, beta-cypermethrin, carbendazim, chlorpyrifos, difenoconazole, cyazofamid, and azoxystrobin. Pyraclostrobin was the most commonly detected pesticide, followed by chlorpyrifos, difenoconazole, and beta-cypermethrin. Pesticide residues in bayberry samples (N = 157) were reported by Yang et al. [30]. A monitoring survey was carried out in China during 2013–2014. A total of 23 pesticides were detected among the 44 pesticides analysed. Only 2.5% of the samples contained pesticide residues (acetamiprid) above the MRL. The most frequently detected pesticide was carbendazim. A study by Yang et al. [31] focused on pesticide contamination of minor fruit (starfruits and Indian jujubes; N = 87), collected from Xiamen, China. Of the total 50 pesticides analysed, 19 residues were found. These researchers found that 67.8% of the samples exceeded the MRLs in minor fruits. The most frequently detected pesticides were carbendazim, followed by imidacloprid, chlorbenzuron, chlorpyrifos, pyridaben, and thiophanate-methyl. Acetamiprid, carbendazim, chlorantraniliprole, chlorpyrifos, diflubenzuron, fenpropathrin, imidacloprid, thiophanate-methyl, and triazophos residues exceeded the MRLs. Pesticide residues in kumquat fruits from China were investigated by Zhang et al. [32] from 2016 to 2020. Among the 573 samples of kumquat, 30 of the 89 targeted pesticides were accumulatively detected. The residues in 9.4% of the samples were higher than the MRLs. The highest detection rates were found for tebuconazole and spirodiclofen, followed by profenofos, cyhalothrin, difenoconazole, imidacloprid, thiophanate-methyl, chlorpyrifos, prochloraz, propargite, carbendazol, and hexythiazox. The residue levels of seven pesticides (avermectin, bifenthrin, difenoconazole, methidathion, profenofos, spirodiclofen, and triazophos) in the kumquat samples surpassed the MRLs. Bouagga et al. [45] conducted a study during 2015–2017, analysing the residues of pesticides in 64 table grape samples from different regions of Tunisia. The presence of 96 pesticides was assessed. A total of 94% of the samples exceeded the MRLs for at least one compound (bifenthrin, carbendazim, carbofuran, chlorpyrifos, deltamethrin, dimethoate, fluazinam, flusilazole, malathion, omethoate, thiacloprid, thiophanate-methyl, tetradifon, and triadimefon). In Egypt, El-Sheikh et al. [46] analysed a total of 120 fruits and vegetables (54 samples of fruits and 66 samples of vegetables) from markets in Sharkia Governorate during the period from July 2020 to June 2021. It was found that 35 out of 90 pesticide residues (38.9%) detected in fruits exceeded the MRLs. Li et al. [47] presented the results from a 2-year monitoring survey (2013–2014) aimed at investigating the presence of pesticide residues in Chinese pears. A total of 43 out of 104 pesticides were detected in 310 samples, of which 2.6% exceeded the MRLs. Acetamiprid, carbendazim, and cyhalothrin were the most frequently detected pesticides. Residues of cyfluthrin, difenoconazole, omethoate, profenofos, pyrimethanil, and tebuconazole exceeded the MRLs. Liu et al. [48] investigated the levels of pesticides in four fruits (Miaoxi yellow peach, lanxi loquat, qingyuan sweet spring tangelo, and Haining pear) in Zhengjiang Province, China, between 2020 and 2021. In this study, 68 pesticides were analysed, of which 25 were detected in the 68 samples. In one Haining pear sample (1.47%), the residue, prochloraz, exceeded the MRL. Pyraclostrobin, prochloraz, carbendazim, and acetamiprid had the highest detection rates in the four fruits. Łozowicka et al. [49] investigated the residue levels of 130 pesticides in raspberry samples (N = 184) from northeastern Poland producers over seven years (2005–2011). Among the 130 analysed pesticides, 18 active substances were found. Pesticide residues exceeding the MRLs were found in 29.3% of samples. Pyrimethalin, fenhexamid, cyprodinil, boscalid, and procymidone were the most frequently detected pesticides. MRL exceedances were found for chlorothalonil, fenazaquin, and procymidone. In another Polish study, Łozowicka et al. [50] focused on pesticide contamination of 392 fruit samples collected from May 2010 to October 2012. Out of 160 pesticides tested, 31 pesticides were found above the limit of detection. In 5.9% of samples, pesticide residues exceeded the MRLs. Dithiocarbamate, captan, cyprodinil, and boscalid were the pesticides most frequently found. The most frequently detected pesticide above the permissible limits were dithiocarbamates, fenazaquin, fenitrothion, esfenwalerate, and flusilazole. Bempah and Donkor [54] analysed a total of 320 fruit samples from the Ghana markets in the Accra Metropolis for almost a year (from September 2007 to June 2008). A total of 16 pesticides (mainly organochlorines) were monitored. The researchers found that 38.6% of the samples contained pesticide residues (endrin aldehyde, heptachlor, lindane, methoxychlor, and p,p’-DDT) above MRLs. In Spain, Berrada et al. [55] investigated pesticide residues in market fruits from one Valencian cooperative. A total of 41 pesticides were monitored. Of the 429 samples, 3.7% of the collected samples exceeded the MRLs. The most frequent residues were dithiocarbamates, fenthion, malathion, chorothalonil, chlorpyrifos, and methidathion. A market-based survey was carried out to evaluate levels of 26 pesticides in fruits in the Hyderabad region, Pakistan, during the period of October 2010–April 2011 [56]. Out of 131 samples analysed, 53 (40.5%) were contaminated with pesticide residues, while only 3 (2.3%) samples exceeded the MRLs for chlorpyrifos. The most frequently detected pesticide was chlorpyrifos, followed by endosulfan sulfate and dieldrin. A total of 312 peach samples were collected from the major producing regions in nine provinces of China from 2013 to 2015 by Li et al. [57]. Out of 105 pesticides analysed, 39 were found. Residues in 3.2% of the monitored samples exceeded their MRLs, involving eight pesticides (carbendazim, cyhalothrin, cypermethrin, difenoconazole, deltamethrin, fenbuconazole, flusilazole, and isazofos). The most frequently detected pesticide was carbendazim, followed by cyhalothrin, chlorpyrifos, acetamiprid, cypermethrin, imidacloprid, difenoconazole, tebuconazole, and pyridaben. In a separate study on raspberry samples, Łozowicka et al. [58] evaluated the residues of 140 pesticides (17 of which were detected) in analysed samples from northeastern Poland (2005–2010). Of the 128 samples, 27 (21.1%) raspberry samples exceeded the MRLs. Pyrimethanil, fenhexamid, cyprodinil, procymidone, and boscalid were the most frequently detected. The levels of detected pesticide residues exceeded the MRLs, only in the cases of chlorothalonil, dithiocarbamates, fenazaquin, and procymidone. Further, in a separate study on apple samples (n = 212), Łozowicka et al. [59] discussed the occurrence of 127 active substances of pesticides (19 of which were detected). The study showed that 7% of the samples from northeastern Poland contained pesticide residues (cyprodinil, dimethoate, diazinon, fenitrothion, flusilazole, and pirimethanil) above the MRLs. Captan, myclobutanil, penconazole, carbendazim, and dithianon were the most commonly used fungicides, while chlorpiryfos, thiachloprid, and fenoxycarb dominated among the insecticides. Also in Poland, Matyaszek et al. [60] determined the incidence of pesticide residues in berries (N = 250) harvested from southeastern Poland in 2009–2011. Of the 148 pesticides tested, 126 were detected. MRLs were exceeded in 4% (cypermetrine) of the samples. The most commonly found active substances included the following: pyrimethanil, dithiocarbamates, procymidone, cypermetrine, cyprodinil, and difenoconazole. Parveen et al. [61] screened 120 samples of fruits for 25 different pesticides during 2008–2009 in Karachi, Pakistan. A total of 22% of samples contained residues of pesticides above the MRLs. Wołejko et al. [62] determined the residues of pesticides in raw and processed berries (N = 170) from Poland (2011). The study included 160 pesticides, of which 29 were detected. In 14.7% of the samples, residues were found above the MRLs. Nine pesticides (acetamiprid, carbendazim, cypermethrin, difenoconazole, dithiocarbamates, esfenvalerate, fenazaquin, flusilazole, and procymidone) exceeding MRL were observed. Boscalid, dithiocarbamates, cyprodinil, carbendazim, pyrimethanil, pyraclostrobin, iprodione, and difenoconazole were the most detected pesticide residues. Zhang et al. [63] developed a multi-residue method for the analysis of 284 pesticides in five local fruit cultivars in Shanghai, China. A total of 58 pesticide residues were detected. The authors reported that pesticide concentration exceeded the MRLs in 5% of the total tested samples (N = 260). The levels of fluopyram residues in 13 samples exceeded the MRL. Thus, the percentage of pesticide residues in fruit samples exceeding the MRL ranged from 1.47% to 94% across various studies.

In addition, pesticide residues above the MRLs were detected in several countries when fruits and vegetables were tested simultaneously. Pesticide residues exceeding the MRLs were detected in fruits and vegetables in the following studies: Algharibeh and AlFararjeh [33] in Jordan (N = 158; 22%), Gondo et al. [34] in Botswana (N = 83; 13%), Hjorth et al. [35] in Nordic countries (N = 724; 8.4%; fruits and vegetables from South America—a Nordic project, monitoring programme), Ibrahim et al. [36] in Egypt (N = 175; 42%), Jallow et al. [37] in Kuwait (N = 150; 21%), Mac Loughlin et al. [38] in Argentina (N = 135; 36.3%), Mebdoua et al. [39] in Algeria (N = 160; 12.5%), Mutengwe et al. [40] in South Africa (N = 53; 1.9%), Poulsen et al. [41] in Denmark [N = 17,309—fruit and vegetable (70%); 2.6% (most frequently in fruit), monitoring programme], Skretteberg et al. [42] in Nordic countries (N = 721; 12%; fruits and vegetables from Southeast Asia—a Nordic project, monitoring programme), Soydan et al. [43] in Turkey (N = 3044; 11.6%), Toptanci et al. [44] in Turkey (N = 493; 29.2%), Jardim and Caldas in Brazil [51] (N = 13,556; 2.7%; Brazilian monitoring programme), Mert et al. [52] in the UK (N = 25,822; 4.0%; monitoring programme), Park et al. [53] in Korea (N = 1146; 1.0%), Al-Shamary et al. [64] in Qatar (N = 127; 62.2%), Bempah et al. [65] in Ghana (N = 350; 19.0%), Chen et al. [66] in China (N = 3009; 11.7%; monitoring programme), Knežević et al. [67] in Croatia (N = 866; 5.3%), Luo et al. [68] in China (N = 3307; 1.0%), Mutengwe et al. [69] in South Africa (N = 199; 1.0%), Mutengwe et al. [70] in South Africa (N = 37,838; 0.32%; monitoring programme), Patiño et al. [71] in Colombia (N = 100; 41.0%), Sivaperumal et al. [72] in India (N = 286; 16.4%), and Szpyrka et al. [73] in Poland (N = 1026; 1.8%). Thus, the percentage of pesticide residues in fruit and vegetable samples exceeding the MRL ranged from 0.32% to 62.2% across various studies. Moreover, the use of banned/withdrawn/non-authorized/restricted/non-registered/non-recommended active ingredients was identified in fruits and vegetables from many different countries [14,15,16,19,20,21,23,24,26,29,30,31,32,33,34,37,38,40,45,48,49,50,51,57,58,59,60,62,69,70,71,73].

In order to enhance the level of food safety for the benefit of consumer protection, the European Union established the Rapid Alert System for Food and Feed (RASFF) in 197. Members of the network use RASFF to notify each other about dangerous health hazards such as heavy metals, pathogenic microorganisms, and pesticide residues. Moreover, RASFF reported pesticides as the third most widely known hazard category [4]. For the same reason, EFSA publishes an annual report on pesticide residues in all food commodities in the European Union every year, based on the results of a surveillance programme involving all Member States [75,76,77,78].

An overview of the number of residue residues per sample is shown in Table 3a,b.

Many samples contained several pesticide residues. A total of 5078 individual pesticide residues were found in the 1354 fruit samples containing residues. Of the 2164 samples analysed, a single pesticide residue was detected in 252 (11.65%) samples, while two, three, four, five, and six pesticide residues were detected in 278 (12.85%), 227 (10.49%), 203 (9.38%), 145 (6.70%), and 103 (4.76%) samples, respectively. Seven or more pesticide residues were detected in 6.75% of the samples. The highest number of pesticide residues was found in two samples of grapefruit, each containing 44 pesticide residues.

In the previously cited studies, multiple pesticide residues (more than two pesticides per sample) were found in most fruit samples: 17.9% in loquats from Lebanon [15], 21.0% in lemons from Turkey [16], 20.0% in fruits from Turkey [17], 22.6% in apples from Lebanon [18], 39.0% in fruits from Egypt [20], 37.1% in table grapes from Turkey [21], 15.9% in strawberry from China [22], 70.0% in citrus fruits from China [23], 82.9% in citrus fruits from Serbia [26], 96.6% in litchies from China [29], 45.6% in Bayberry from China [30], 55.2% in minor fruit from China [31], 80.8% in Kumquats from China [32], 100% in table grapes from Tunisia [45], 69.7% in pears from China [47], 19.1% in fruits from China [48], 37.5% in raspberries from Poland [49], 30.1% in fruits Poland [50], 14.9% in fruits from Spain [55], 76.3% in peaches from China [57], 37.5% in raspberries from Poland [58], 51 and 34.1% in berries from Poland [60,62], and 56.2% in fruits from China [63]. In addition, multiple pesticide residues were found in fruits from the United Arab Emirates [25], Mexico [28], Egypt [46], Poland [59], and Pakistan [61].

## 4. Conclusions

In this study, pesticide residues were determined in 2164 samples of the most popularly consumed fruits in Serbia (during 2016–2019). There were 1354 (62.57%) samples contaminated with pesticide residues, of which 101 (4.67%) samples exceeded the MRLs. Among the 101 fruit samples with MRL exceedances, pomegranate was the fruit with the highest number of MRL exceedances (26 samples, 25.74%), followed by grapefruit with 21 samples (20.79%), mandarin with 14 samples (13.86%), apple with 12 samples (11.88%), orange with 7 samples (6.93%), pear with 6 samples (5.94%), lemon and lime with 4 samples each (3.96%), pineapple with 3 samples (2.97%), grapes with 2 samples (1.98%), and avocado and sour cherry with 1 sample each (0.99%). Pomegranate, grapefruit, and mandarin showed the highest number of samples with multiple pesticide residues above the MRLs. These results highlight the need to continuously monitor pesticide residues in fruits in order to fully protect public health.

## Figures and Tables

**Figure 1 foods-14-01828-f001:**
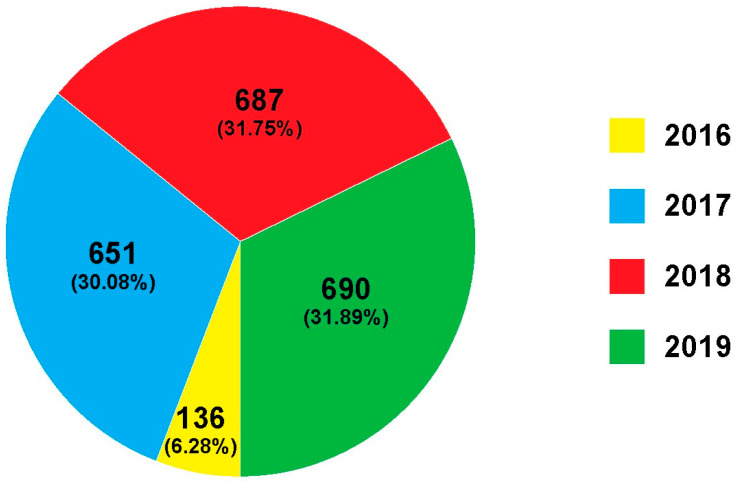
Number and percentage of samples analysed in 2016, 2017, 2018, and 2019.

**Figure 2 foods-14-01828-f002:**
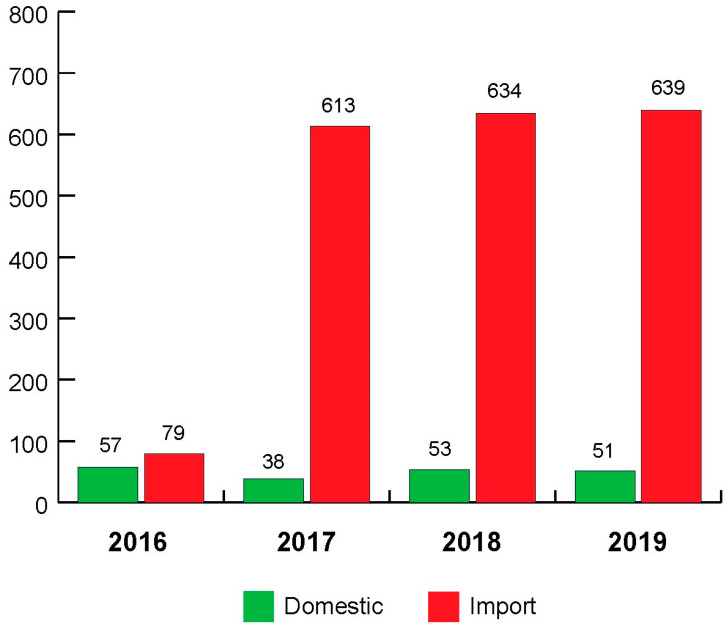
Origin of samples analysed in 2016, 2017, 2018, and 2019.

**Table 1 foods-14-01828-t001:** Characteristics of the analysed fruit samples and number of fruit samples without and with pesticide residues.

Name of the Fruit Samples	Country of Origin	No. of Samples	No. of Samples Without Residues (<0.01 mg/kg)	%	No. of Samples with Residues At or Above 0.01 mg/kg	%	No. of Samples with Residues Above the MRL	%
Almond	Spain (N = 1), United States of America (N = 5)	6	5	83.33	1	16.67	0	0
Apple	Albania (N = 11), Austria (N = 3), Bosnia and Herzegovina (N = 3), Bulgaria (N = 1), Chile (N = 1), Croatia (N = 13), France (N = 1), Greece (N = 4), Hungary (N = 3), Italy (N = 32), North Macedonia (N = 12), Poland (N = 171), Serbia (N = 88), Slovenia (N = 2), The Netherlands (N = 4), Turkey (N = 2)	351	126	35.90	225	64.10	12	3.42
Apricot	Bulgaria (N = 1), Greece (N = 4), Italy (N = 2), Serbia (N = 6), Spain (N = 2)	15	10	66.67	5	33.33	0	0
Aronia	Serbia (N = 1)	1	1	100	0	0	0	0
Avocado	Chile (N = 1), Colombia (N = 2), Israel (N = 1), Kenya (N = 4), Peru (N = 6), South Africa (N = 1), Tanzania (N = 2), Turkey (N = 1), Zimbabwe (N = 1)	19	12	63.16	7	36.84	1	5.26
Banana	Brazil (N = 2), Colombia (N = 41), Costa Rica (N = 35), Dominican Republic (N = 1), Ecuador (N = 47), Ghana (N = 1), Guatemala (N = 8), Honduras (N = 7), Italy (N = 1), Mexico (N = 19), Panama (N = 6)	168	70	41.67	98	58.33	0	0
Blackberry	Serbia (N = 2)	2	1	50.00	1	50.00	0	0
Blueberry	Argentina (N = 1), Serbia (N = 1), Spain (N = 1), Peru (N = 1), The Netherlands (N = 1)	5	4	80.00	1	20.00	0	0
Brazil nut	Bolivia (N = 2)	2	2	100	0	0	0	0
Carambola	Malesia (N = 2)	2	2	100	0	0	0	0
Cashew	Vietnam (N = 1)	1	1	100	0	0	0	0
Chestnut	China (N = 1), Serbia (N = 1)	2	2	100	0	0	0	0
Clementine	Egypt (N = 1), Italy (N = 5), Spain (N = 7), Turkey (N = 3)	16	8	50.00	8	50.00	0	0
Coconut	Ivory Coast (N = 1)	1	0	0	1	100	0	0
Currants	The Netherlands (N = 2)	2	0		2	100	0	0
Date palm	Iran (N = 7), Israel (N = 1)	8	6	75.00	2	25.00	0	0
Fig	Turkey (N = 1)	1	1	100	0	0	0	0
Grapefruit	Cyprus (N = 2), Greece (N = 3), Israel (N = 1), Mexico (N = 2), South Africa (N = 33), Swaziland (N = 1), Turkey (N = 71), Zimbabwe (N = 2)	115	13	11.30	102	88.70	21	18.26
Grape	Bosnia and Herzegovina (N = 2), Chile (N = 1), Greece (N = 1), India (N = 2), Italy (N = 18), North Macedonia (N = 111), Serbia (N = 8), South Africa (N = 1), Turkey (N = 8)	152	99	65.13	53	34.87	2	1.32
Hazelnut	Croatia (N = 3), Georgia (N = 1), Turkey (N = 1)	5	4	80.00	1	20.00	0	0
Japanese apple	Albania (N = 1), Spain (N = 8)	9	8	88.89	1	11.11	0	0
Kiwi	Chile (N = 7), Greece (N = 21), Italy (N = 16), New Zealand (N = 4), North Macedonia (N = 2)	50	33	66.00	17	34.00	0	0
Kumquat	Israel (N = 1), Italy (N = 2), South Africa (N = 1)	4	2	50.00	2	50.00	0	0
Lemon	Argentina (N = 45), Egypt (N = 1), Greece (N = 12), Italy (N = 2), South Africa (N = 15), Spain (N = 32), Turkey (N = 107), Uruguay (N = 3)	217	42	19.35	175	80.65	4	1.84
Lime	Brazil (N = 1), China (N = 1), Guatemala (N = 2), Mexico (N = 21), South Africa (N = 1), The Netherlands (N = 1), Turkey (N = 1)	28	4	14.29	24	85.71	4	14.29
Lychee	Chile (N = 1)	1	1	100	0	0	0	0
Mandarin	Albania (N = 4), Croatia (N = 14), Cyprus (N = 4), Egypt (N = 2), Greece (N = 42), Italy (N = 3), Morocco (N = 1), Spain (N = 12), Swaziland (N = 1), Turkey (N = 111)	194	60	30.93	134	69.07	14	7.22
Mango	Brazil (N = 9), Burkina Faso (N = 1), Dominican Republic (N = 1), Israel (N = 1), Ivory Coast (N = 1), Mali (N = 2), Peru (N = 7), Senegal (N = 2)	24	15	62.50	9	37.50	0	0
Orange	Egypt (N = 32), Greece (N = 125), Morocco (N = 9), Italy (N = 3), South Africa (N = 46), Spain (N = 38), The Netherlands (N = 1), Turkey (N = 55), Uruguay (N = 3), Zimbabwe (N = 12)	324	98	30.25	226	69.75	7	2.16
Passion fruit	South Africa (N = 1)	1	1	100	0	0	0	0
Peach	France (N = 1), Greece (N = 32), Italy (N = 3), Serbia (N = 13), Spain (N = 7)	56	32	57.14	24	42.86	0	0
Peach (nectarine)	Albania (N = 2), Belgium (N = 1), Greece (N = 14), Italy (N = 4), North Macedonia (N = 3), Serbia (N = 5), Spain (N = 7)	36	23	63.89	13	36.11	0	0
Peanut	Argentina (N = 4), China (N = 4)	8	7	87.50	1	12.50	0	0
Pear	Argentina (N = 2), Belgium (N = 6), Bosnia and Herzegovina (N = 2), China (N = 3), Greece (N = 2), Italy (N = 9), Poland (N = 9), Serbia (N = 30), South Africa (N = 4), Spain (N = 3), The Netherlands (N = 24), Turkey (N = 1)	95	27	28.42	68	71.58	6	6.32
Pineapple	Colombia (N = 13), Costa Rica (N = 25), Ecuador (N = 1), Italy (N = 1), Ivory Coast (N = 1)	41	11	26.83	30	73.17	3	7.32
Pistachio	Turkey (N = 1)	1	1	100	0	0	0	0
Pitaya	Thailand (N = 1), Vietnam (N = 1)	2	2	100	0	0	0	0
Plum	Albania (N = 3), Greece (N = 3), Italy (N = 2), Moldova (N = 1), North Macedonia (N = 3), Serbia (N = 14)	26	15	57.69	11	42.31	0	0
Pomegranate	Argentina (N = 2), Egypt (N = 1), Greece (N = 7), Peru (N = 6), Turkey (N = 50)	66	24	36.36	42	63.64	26	39.39
Pomelo	China (N = 12)	12	4	33.33	8	66.67	0	0
Quince	Greece (N = 1)	1	1	100	0	0	0	0
Raspberry	Morocco (N = 2), Serbia (N = 3), Spain (N = 2), The Netherlands (N = 1)	8	4	50.00	4	50.00	0	0
Sour cherry	Hungary (N = 8), Serbia (N = 6)	14	8	57.14	6	42.86	1	7.14
Strawberry	Albania (N = 8), Germany (N = 1), Greece (N = 13), Poland (N = 3), Serbia (N = 15), Spain (N = 5), Turkey (N = 1)	46	11	23.91	35	76.09	0	0
Sweet cherry	Greece (N = 6), Hungary (N = 1), Italy (N = 1), North Macedonia (N = 1), Poland (N = 1), Romania (N = 3), Serbia (N = 6), Spain (N = 1)	20	4	20.00	16	80.00	0	0
Walnut	Bulgaria (N = 2), Russia (N = 2), Ukraine (N = 1), United States of America (N = 1)	6	5	83.33	1	16.67	0	0

**Table 2 foods-14-01828-t002:** The frequency of the detected pesticide residues and their concentrations in fruit samples.

Pesticide Name (N = 173)	Types of Pesticide	Frequency of Detection in 2164 Samples	%	No. of Samples with Residues Above MRL	%	Range Min–Max (mg/kg)
2-Phenylphenol	Fungicide	44	2.03	0	0	0.010–7.028
Abamectin	Insecticide	2	0.09	0	0	0.010
Acephate	Insecticide	5	0.23	0	0	0.010–0.011
Acetamiprid	Insecticide	171	7.90	21 (apple; grapefruit, N = 2; mandarin; pomegranate, N = 17)	12.21	0.010–1.418
Acetochlor	Herbicide	2	0.09	0	0	0.010
Acibenzolar-S-methyl	Fungicide	31	1.43	4 (grapefruit, N = 3; mandarin)	12.90	0.010–0.114
Acrinathrin	Insecticide	2	0.09	0	0	0.011–0.020
Aldicarb	Insecticide	27	1.25	0	0	0.010–0.018
Aldicarb sulfone	Insecticide	2	0.09	0	0	0.010
Ametryn	Herbicide	12	0.55	0	0	0.010
Amitraz	Insecticide	58	2.68	5 (pomegranate)	9	0.010–1.339
Atrazine	Herbicide	10	0.46	0	0	0.010–0.018
Azinphos-ethyl	Insecticide	6	0.28	1 (grapefruit)	16.67	0.010–0.982
Azinphos-methyl	Insecticide	7	0.32	0	0	0.010–0.042
Azoxystrobin	Fungicide	96	4.44	0	0	0.010–0.717
Bendiocarb	Insecticide	3	0.14	0	0	0.010
Bifenazate	Insecticide	6	0.28	0	0	0.011–0.020
Bifenthrin	Insecticide	18	0.83	0	0	0.010–0.081
Biphenyl	Fungicide	1	0.05	0	0	0.010
Bitertanol	Fungicide	5	0.23	1 (avocado)	20.00	0.012–0.152
Boscalid	Fungicide	154	7.12	1 (pomegranate)	0.65	0.010–2.989
Buprofezin	Insecticide	79	3.65	0	0	0.010–0.589
Butachlor	Herbicide	1	0.05	0	0	0.012
Butoxycarboxim	Insecticide	5	0.23	5 (grapefruit, N = 2; pomegranate, N = 3)	100	0.034–0.037
Carbaryl	Insecticide	29	1.34	0	0	0.010
Carbendazim	Fungicide	185	8.55	15 (apple, N = 3; grapefruit; lemon; orange, N = 2; pear, n = 2; pomegranate, N = 6)	8.06	0.010–2.670
Carbofuran	Insecticide	1	0.05	1 (mandarin)	100	0.033
Carboxin	Fungicide	2	0.09	0	0	0.010
Carfentrazone-ethyl	Herbicide	6	0.28	0	0	0.010–0.012
Chlorantraniliprole	Insecticide	24	1.11	0	0	0.010–0.121
Chlorothalonil	Fungicide	5	0.23	1 (grape)	20.00	0.010–12.500
Chlorotoluron	Herbicide	22	1.02	8 (grapefruit, N = 2; mandarin; pomegranate, N = 5)	36.36	0.010–0.928
Chlorpropham	Herbicide	2	0.09	0	0	0.010
Chlorpyrifos	Insecticide	145	6.70	10 (grapefruit, N = 8; grape; pomegranate)	6.90	0.010–2.338
Chlorpyrifos-methyl	Insecticide	27	1.25	1 (grapefruit)	3.70	0.011–0.944
Clethodim	Herbicide	1	0.05	0	0	0.010
Clofentezine	Insecticide	5	0.23	0	0	0.011–0.084
Clothianidin	Insecticide	13	0.60	0	0	0.010–0.045
Cyazofamid	Fungicide	3	0.14	0	0	0.054–0.111
Cyfluthrin	Insecticide	1	0.05	0	0	0.012
Cymoxanil	Fungicide	8	0.37	0	0	0.010–0.023
Cypermethrin	Insecticide	10	0.46	2 (pomegranate)	18.18	0.011–0.301
Cyproconazole	Fungicide	2	0.09	0	0	0.010–0.016
Cyprodinil	Fungicide	38	1.76	7 (mandarin; pomegranate, N = 6)	18.42	0.011–0.606
Deltamethrin	Insecticide	20	0.92	12 (grapefruit; mandarin; pomegranate, N = 10)	60.00	0.010–0.326
Diazinon	Insecticide	3	0.14	0	0	0.010
Dicrotophos	Insecticide	1	0.05	0	0	0.010
Difenoconazole	Fungicide	30	1.39	2 (pomegranate)	6.67	0.010–0.174
Diflubenzuron	Insecticide	8	0.37	2 (pear)	25.00	0.010–0.073
Dimethoate	Insecticide	8	0.37	1 (apple)	12.50	0.010–0.047
Dimethomorph	Fungicide	65	3.00	1 (grapefruit)	1.54	0.010–0.737
Dimoxystrobin	Fungicide	8	0.37	0	0	0.010
Dinotefuran	Insecticide	10	0.46	1 (pomegranate)	10.00	0.010–0.038
Diphenylamine	Fungicide	1	0.05	0	0	0.010
Emamectin	Insecticide	2	0.09	0		0.020–0.044
Emamectin B1a	Insecticide	4	0.18	0	0	0.010–0.025
Emamectin B1b	Insecticide	5	0.23	1 (pear)	20.00	0.010–0.016
Eprinomectin	Insecticide	2	0.09	0	0	0.010
Ethiofencarb	Insecticide	12	0.55	3 (grapefruit)	25.00	0.010–0.069
Ethirimol	Fungicide	5	0.23	0	0	0.010–0.038
Ethofumesate	Herbicide	20	0.92	0	0	0.010–0.038
Etofenprox	Insecticide	1	0.05	0	0	0.141
Etoxazole	Insecticide	19	0.88	1 (pomegranate)	5.26	0.010–0.024
Famoxadone	Fungicide	3	0.14	0	0	0.040–0.292
Fenamidone	Fungicide	1	0.05	0	0	0.042
Fenamiphos	Insecticide	3	0.14	0	0	0.012–0.016
Fenazaquin	Insecticide	3	0.14	0	0	0.010
Fenbuconazole	Fungicide	4	0.18	0	0	0.010–0.033
Fenhexamid	Fungicide	18	0.83	0	0	0.010–0.645
Fenoxycarb	Insecticide	1	0.05	0	0	0.013
Fenpropimorph	Fungicide	7	0.32	0	0	0.010–0.067
Fenpyroximate	Insecticide	5	0.23	0	0	0.010–0.047
Fenthion	Insecticide	1	0.05	0	0	0.010
Fenuron	Herbicide	2	0.09	0	0	0.010
Fenvalerate	Insecticide	51	2.36	31 (lemon; mandarin, N = 9; orange, N = 3; pomegranate, N = 18)	60.78	0.010–1.247
Flonicamid	Insecticide	9	0.42	0	0	0.010–0.086
Fluazifop-butyl	Herbicide	1	0.05	0	0	0.010
Fludioxonil	Fungicide	154	7.12	2 (pear; pineapple)	1.30	0.010–5.984
Flufenacet	Herbicide	3	0.14	0	0	0.021–0.035
Flufenoxuron	Insecticide	2	0.09	0	0	0.015–0.032
Fluopyram	Fungicide	2	0.09	0	0	0.068–0.152
Fluoxastrobin	Fungicide	4	0.18	0	0	0.010–0.018
Flutolanil	Fungicide	2	0.09	0	0	0.010
Flutriafol	Fungicide	7	0.32	0	0	0.013–0.039
Formothion	Insecticide	17	0.79	10 (apple, N = 8; mandarin, N = 2)	58.82	0.010–0.396
Hexaconazole	Fungicide	5	0.23	0	0	0.010
Hexythiazox	Insecticide	5	0.23	0	0	0.013–0.032
Imazalil	Fungicide	624	28.84	10 (grapefruit, N = 3; lemon, N = 2; mandarin, N = 2; orange, N = 2; pomegranate)	1.60	0.010–93.349
Imidacloprid	Insecticide	213	9.84	0	0	0.010–0.327
Indoxacarb	Insecticide	24	1.11	4 (pomegranate)	16.67	0.010–0.080
Ipconazole	Fungicide	1	0.05	0	0	0.010
Iprodione	Fungicide	3	0.14	2 (orange)	66.67	0.019–0.551
Iprovalicarb	Fungicide	2	0.09	0	0	0.011–0.050
Isoprocarb	Insecticide	1	0.05	0	0	0.010
Isoproturon	Herbicide	1	0.05	0	0	0.010
Ketoconazole	Fungicide	19	0.88	0	0	0.010
Kresoxim-methyl	Fungicide	4	0.18	0	0	0.013–0.047
Lambda-cyhalothrin	Insecticide	8	0.37	0	0	0.010–0.098
Lufenuron	Insecticide	60	2.77	2 (grapefruit; pomegranate)	3.33	0.011–0.787
Malaoxon	Insecticide	3	0.14	0	0	0.058–1.067
Malathion	Insecticide	22	1.02	0	0	0.010–0.707
Mandipropamid	Fungicide	4	0.18	0	0	0.038–0.655
Mepanipyrim	Fungicide	13	0.60	8 (grapefruit, N = 3; mandarin; pomegranate, N = 4)	61.54	0.010–0.103
Mepronil	Fungicide	1	0.05	1 (grapefruit)	100	0.026
Metaflumizone	Insecticide	3	0.14	0	0	0.010–0.050
Metalaxyl	Fungicide	31	1.43	0	0	0.010–0.580
Metalaxyl-M	Fungicide	10	0.46	0	0	0.010–0.347
Methabenzthiazuron	Herbicide	5	0.23	0	0	0.010
Methamidophos	Insecticide	36	1.66	15 (grapefruit, N = 3; mandarin; orange; pear; pomegranate, N = 9)	41.67	0.010–2.048
Methidathion	Insecticide	1	0.05	0	0	0.012
Methiocarb	Insecticide	12	0.55	0	0	0.010–0.176
Methomyl	Insecticide	52	2.40	3 (grapefruit)	5.77	0.010–0.593
Methoxyfenozide	Insecticide	94	4.34	0	0	0.010–1.129
Metobromuron	Herbicide	5	0.23	2 (grapefruit)	40.00	0.010–0.598
Metrafenone	Fungicide	1	0.05	0	0	0.051
Metribuzin	Herbicide	42	1.94	2 (grapefruit)	4.76	0.010–0.335
Monocrotophos	Insecticide	3	0.14	0	0	0.010
Myclobutanil	Fungicide	34	1.57	0	0	0.011–0.290
Nitenpyram	Insecticide	1	0.05	0	0	0.010
Novaluron	Insecticide	7	0.32	2 (mandarin; orange)	28.57	0.010–0.013
Nuarimol	Fungicide	3	0.14	2 (grapefruit; mandarin)	66.67	0.010–0.043
Omethoate	Insecticide	4	0.18	1 (sour cherry)	25.00	0.010–0.060
Oxadixyl	Fungicide	13	0.60	6 (grapefruit, N = 4; mandarin; pomegranate)	46.15	0.010–0.595
Oxamyl	Insecticide	10	0.46	10 (grapefruit; lime, N = 4; mandarin; pomegranate, N = 4)	100	0.038–3.161
Penconazole	Fungicide	18	0.83	0	0	0.011–0.175
Permethrin	Insecticide	1	0.05	0	0	0.020
Phenmedipham	Herbicide	1	0.05	0	0	0.010
Phosmet	Insecticide	10	0.46	0	0	0.010–0.284
Picoxystrobin	Fungicide	43	1.99	19 (grapefruit, N = 16; mandarin; pomegranate, N = 2)	44.19	0.010–2.439
Piperonyl-butoxide	Insecticide	2	0.09	1 (pineapple)	50.00	0.010–0.130
Pirimicarb	Insecticide	32	1.48	0	0	0.010–0.110
Pirimiphos-methyl	Insecticide	12	0.55	10 (grapefruit, N = 3; mandarin, N = 2; pomegranate, N = 5)	83.33	0.010–0.480
Prochloraz	Fungicide	143	6.61	2 (pomegranate; sour cherry)	1.40	0.010–3.905
Promecarb	Insecticide	7	0.32	5 (grapefruit, N = 2; mandarin; pomegranate, N = 2)	71.43	0.010–0.039
Prometon	Herbicide	4	0.18	0	0	0.010
Prometryn	Herbicide	4	0.18	3 (grapefruit)	75.00	0.010–0.106
Propargite	Insecticide	4	0.18	1 (grapefruit)	25.00	0.010–0.316
Propham	Herbicide	59	2.73	3 (lime; pomegranate, N = 2)	5.08	0.010–0.085
Propiconazole	Fungicide	113	5.22	3 (pomegranate)	2.65	0.010–3.143
Propoxur	Insecticide	32	1.48	2 (pomegranate)	6.25	0.010–0.127
Prothioconazole	Fungicide	97	4.48	5 (grapefruit, N = 3; mandarin, N = 2)	5.15	0.010–0.586
Pymetrozine	Insecticide	2	0.09	0	0	0.010–0.019
Pyracarbolid	Fungicide	5	0.23	1 (grapefruit)	20.00	0.012–0.017
Pyraclostrobin	Fungicide	93	4.30	0	0	0.010–0.153
Pyridaben	Insecticide	23	1.06	0	0	0.010–0.135
Pyrimethanil	Fungicide	245	11.32	2 (pomegranate)	0.82	0.010–6.633
Pyriproxyfen	Insecticide	138	6.38	0	0	0.010–0.150
Quizalofop-p-ethyl	Herbicide	1	0.05	0	0	0.034
Siduron	Herbicide	1	0.05	0	0	0.010
Spinetoram B	Insecticide	2	0.09	0	0	0.014–0.017
Spirodiclofen	Insecticide	16	0.74	0	0	0.010–0.195
Spiromesifen	Insecticide	29	1.34	9 (grapefruit, N = 3; mandarin; pomegranate, N = 5)	31.03	0.010–0.464
Spirotetramat	Insecticide	14	0.65	0	0	0.010–0.226
Spiroxamine	Fungicide	9	0.42	0	0	0.010–0.181
Sulfentrazone	Herbicide	4	0.18	2 (grapefruit)	50.00	0.010–0.026
Tebuconazole	Fungicide	116	5.36	1 (pineapple)	0.86	0.010–1.000
Tebufenozide	Insecticide	34	1.57	0	0	0.010–0.051
Tebufenpyrad	Insecticide	9	0.42	0	0	0.010–0.061
Tebuthiuron	Herbicide	11	0.51	7 (grapefruit, N = 2; mandarin; pomegranate, N = 4)	63.64	0.010–0.079
Teflubenzuron	Insecticide	50	2.31	0	0	0.010–0.025
Terbutryn	Herbicide	6	0.28	3 (grapefruit)	50.00	0.010–0.044
Tetraconazole	Fungicide	17	0.79	0	0	0.010–0.065
Thiabendazole	Fungicide	337	15.57	0	0	0.010–4.814
Thiacloprid	Insecticide	63	2.91	2 (grapefruit; pomegranate)	3.17	0.010–0.154
Thiamethoxam	Insecticide	16	0.74	0	0	0.010–0.034
Thiophanate-methyl	Fungicide	32	1.48	0	0	0.010–0.683
Triadimefon	Herbicide	1	0.05	0	0	0.016
Triadimenol	Fungicide	3	0.14	0	0	0.010–0.028
Tricyclazole	Fungicide	3	0.14	3 (grapefruit)	100	0.049–0.074
Trifloxystrobin	Fungicide	24	1.11	2 (pomegranate)	8.33	0.010–0.144
Triflumuron	Insecticide	2	0.09	0	0	0.022–0.048
Triticonazole	Fungicide	3	0.14	0	0	0.010
Zoxamide	Fungicide	7	0.32	3 (grapefruit)	42.86	0.014–1.094

MRL, maximum residue level.

**Table 3 foods-14-01828-t003:** (**a**) Number of pesticide residues in an individual sample. (**b**) Number of pesticide residues in an individual sample.

(**a**)												
No. of pesticide residues	0	1	2	3	4	5	6	7	8	9	10	11
No. of samples	810	252	278	227	203	145	103	65	34	19	7	2
%	37.43	11.65	12.85	10.49	9.38	6.70	4.76	3.00	1.57	0.88	0.32	0.09
No. of samples with residues above the MRL	0	7 (7 with 1)	17 (13 with 1; 4 with 2)	10 (7 with 1; 3 with 2)	12 (4 with 1; 6 with 2; 1 with 3; 1 with 4)	8 (3 with 1; 2 with 2; 2 with 3; 1 with 5)	15 (7 with 1; 6 with 2; 1 with 3; 1 with 4)	7 (3 with 1; 1 with 2; 2 with 3; 1 with 5)	8 (3 with 1; 3 with 2; 1 with 3; 1 with 4)	5 (3 with 1; 1 with 2; 1 with 4)	1 (1 with 1)	0
(**b**)												
No. of pesticide residues	12	13	15	17	26	29	30	31	33	36	39	44
No. of samples	4	4	1	1	1	1	1	1	1	1	1	2
%	0.18	0.18	0.05	0.05	0.05	0.05	0.05	0.05	0.05	0.05	0.05	0.09
No. of samples with residues above the MRL	2 (1 with 1; 1 with 2)	0	0	0	1 (1 with 16)	1 (1 with 17)	1 (1 with 11)	1 (1 with 17)	1 (1 with 18)	1 (1 with 16)	1 (1 with 20)	1 (1 with 19; 1 with 22)

## Data Availability

The original contributions presented in this study are included in the article/Appendix A. Further inquiries can be directed to the corresponding author.

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
