# Peer review of "Determination of Pesticide Residues in Fresh Fruits in the Serbian Market by LC-MS/MS"

_foods, 2025, doi:10.3390/foods14101828_

Round 1

Reviewer 1 Report (Previous Reviewer 2)

Comments and Suggestions for Authors

The authors have significantly improved the manuscript compared to the previous version. They have mostly addressed this reviewers previous concerns and have done exceptionally at improving the introduction, discussion, and references. I recommend accepting for publication following minor revision.

Minor point:

1. Line 120-123: The authors should provide details on the LC MS method. What were the mobile phases? MS settings - voltages, gas flows, etc. 

Author Response

Dear Reviewer,

Thank you very much for your comments and professional advice. These opinions help to improve our article. Based on your suggestion and request, we have made corrected modifications on the manuscript. Furthermore, we would like to show the details as follows:

Comment 1: Line 120-123: The authors should provide details on the LC MS method. What were the mobile phases? MS settings - voltages, gas flows, etc.

Response 1: We inserted details on the LC MS method.

Reviewer 2 Report (Previous Reviewer 1)

Comments and Suggestions for Authors

Dear authors

1) The article has been significantly improved and I think I can recommend it for publication in the journal Foods. However, the authors did not address my question at all regarding the significant (several years) delay between the sample collection period and the article submission (i.e. 2016-2019 and 2024/2025). Perhaps this was due to the lengthy research process or the need to obtain funding for laboratory testing. It is worth noting at this point that in the current science, articles presenting results more than five years old may no longer be considered and therefore cited by other authors.
2) I recommend reducing the font and spacing in Table 1 and Table 2. Also, since both of these tables take up more than one page, the phrase ‘Table 1. Cont.’, ‘Table 2. Cont.’ should be included at the beginning of each subsequent page, followed by repeated headings, otherwise the tables are unreadable.

Author Response

Dear Reviewer,

Thank you very much for your comments and professional advice. These opinions help to improve our article. Based on your suggestion and request, we have made corrected modifications on the manuscript. Furthermore, we would like to show the details as follows:

Comment 1: The article has been significantly improved and I think I can recommend it for publication in the journal Foods. However, the authors did not address my question at all regarding the significant (several years) delay between the sample collection period and the article submission (i.e. 2016-2019 and 2024/2025). Perhaps this was due to the lengthy research process or the need to obtain funding for laboratory testing. It is worth noting at this point that in the current science, articles presenting results more than five years old may no longer be considered and therefore cited by other authors.

Response 1: We fully agree with your comment.

Comment 2: I recommend reducing the font and spacing in Table 1 and Table 2. Also, since both of these tables take up more than one page, the phrase ‘Table 1. Cont.’, ‘Table 2. Cont.’ should be included at the beginning of each subsequent page, followed by repeated headings, otherwise the tables are unreadable.

Response 2: We have rearranged the tables according to your suggestions.

-------

This manuscript is a resubmission of an earlier submission. The following is a list of the peer review reports and author responses from that submission.

Round 1

Reviewer 1 Report

Comments and Suggestions for Authors

The issue raised is of interest because of the significant health risks of consuming food containing pesticide residues, and appropriate to the scope of the Journal “Foods”. The large number of samples tested is noteworthy, but the authors should clarify whether there was a scheme for the selection of fruits and the number of samples taken, and why they have only now decided to publish the results of the study, as it was conducted quite a long time ago, between 2016 and 2019. This information is, moreover, missing from the article and should be added in the “Materials and Methods” section, i.e.: “Over 4-year period (2016-2019)”.

The title of the article is misleading, as it indicates that the fruits examined came from the Republic of Serbia, while they originated from 59 countries. I therefore suggest using the phrase “in the Serbian market” instead of “in the Republic of Serbia”.

The “Materials and Methods” section ends with the phrase “Kecojević et al. [19].”. The sentence “The results were… regulation belongs to this section as well.

Meanwhile, the “Results” section should start with a citation of Table 1, and then the following sentences can be included “A total of 173 (Table 2).... and 20 mg/kg.”. I believe, of course, that both Table 1 and Table 2 should be placed in the “Results” section.

Unfortunately, there was virtually no discussion provided by the Authors, only the results of the study were presented. Therefore, I believe that the “Results” section should be a separate section (as I suggested earlier) and it should be followed by a “Discussion” section citing studies on the presence of pesticide residues in fruits. Here they can use the sources they cite in Supplementary Material 1, which have not, I think, been submitted, and then, of course, no longer include these Materials.

I also include below my editorial comments:

31-32 “They protect or increase yields and may” > “They help to protect crops and grow more yields, their use can also…”

45 EFSA and the Commission. > the European Food Safety Authority (EFSA) and the European Commission.

52 Supplementary material 1 Table S1 > Supplementary Material 1 in Table S1.

Also, it seems to me that only Supplementary Material 2 is included in the submission and Supplementary Material 1 is missing. It should therefore be added.

54 LOQ > “limit of quantification (LOQ)”

59 hazard > hazards

60 fruits (Table 1) > fruits

66 fruits (Table 1) > fruits

70 The limit of quantitation (LOQ) > The LOQ

Second line of Table 1: 4+1≠6. Please also check all other lines and correct if necessary.

85 reported > presented

94 shown (Table 1) that > shown (Table 1) that (Table 1 was already referred to in the previous sentence.)

126 samples (Table 2). > samples. (Table 2 was already referred to in the previous sentence.)

152-158 “Many samples… residues.” (These sentences should be moved below Table 3b.)

Author Response

Dear Reviewer,

Thank you very much for your comments and professional advice. These opinions help to improve our article. Based on your suggestion and request, we have made corrected modifications on the manuscript. Furthermore, we would like to show the details as follows:

Comment 1: The issue raised is of interest because of the significant health risks of consuming food containing pesticide residues, and appropriate to the scope of the Journal “Foods”. The large number of samples tested is noteworthy, but the authors should clarify whether there was a scheme for the selection of fruits and the number of samples taken, and why they have only now decided to publish the results of the study, as it was conducted quite a long time ago, between 2016 and 2019. This information is, moreover, missing from the article and should be added in the “Materials and Methods” section, i.e.: “Over 4-year period (2016-2019)”.

Response 1: Done [we inserted: “Over 4-year period (2016-2019)”].

Comment 2: The title of the article is misleading, as it indicates that the fruits examined came from the Republic of Serbia, while they originated from 59 countries. I therefore suggest using the phrase “in the Serbian market” instead of “in the Republic of Serbia”.

Response 2: Done (the phrase “in the Serbian market” is inserted).

Comment 3: The “Materials and Methods” section ends with the phrase “Kecojević et al. [19].”. The sentence “The results were… regulation belongs to this section as well.

Response 3: Done (the sentence “The results were…” is inserted after the phrase “Kecojević et al. [19].”).

Comment 4: Meanwhile, the “Results” section should start with a citation of Table 1, and then the following sentences can be included “A total of 173 (Table 2).... and 20 mg/kg.”. I believe, of course, that both Table 1 and Table 2 should be placed in the “Results” section.

Response 4: Done (both Tables are moved to the Results section).

Comment 5: Unfortunately, there was virtually no discussion provided by the Authors, only the results of the study were presented. Therefore, I believe that the “Results” section should be a separate section (as I suggested earlier) and it should be followed by a “Discussion” section citing studies on the presence of pesticide residues in fruits. Here they can use the sources they cite in Supplementary Material 1, which have not, I think, been submitted, and then, of course, no longer include these Materials.

Response 5: Done (In the section ’Results and discussion’ we cited studies on the presence of pesticide residues in fruits).

Comment 6: 31-32 “They protect or increase yields and may” > “They help to protect crops and grow more yields, their use can also…”

Response 6: Done (“They protect or increase yields and may” is changed with “They help to protect crops and grow more yields, their use can also…”).

Comment 7: 45 EFSA and the Commission. > the European Food Safety Authority (EFSA) and the European Commission.

Response 7: Done [ “EFSA and the Commission.” is changed with “the European Food Safety Authority (EFSA) and the European Commission.” ].

Comment 8: 52 Supplementary material 1 Table S1 > Supplementary Material 1 in Table S1.

Response 8: Done (“Supplementary material 1 Table S1” is changed with “Supplementary Material 1 in Table S1”).

Comment 9: Also, it seems to me that only Supplementary Material 2 is included in the submission and Supplementary Material 1 is missing. It should therefore be added.

Response 9: Both Supplementary Materials were previously submitted to the Journal. We will again send to the journal both Supplementary Materials.

Comment 10: 54 LOQ > “limit of quantification (LOQ)”

Response 10: Done [“LOQ” is changed with “limit of quantification (LOQ)”].

Comment 11: 59 hazard > hazards

Response 11: Done (“hazard” is changed with “hazards”).

Comment 12: 60 fruits (Table 1) > fruits

Response 12: Done [“fruits (Table 1)” is changed with “fruits”].

Comment 13: 66 fruits (Table 1) > fruits

Response 13: Done [“fruits (Table 1)” is changed with “fruits”].

Comment 14: 70 The limit of quantitation (LOQ) > The LOQ

Response 14: Done [“The limit of quantitation (LOQ)” is changed with “The LOQ”].

Comment 15: Second line of Table 1: 4+1≠6. Please also check all other lines and correct if necessary.

Response 15: Done (correct number is 1+5=6). Also, we found one more mistake, in the section where Japanese apple is mentioned.

Comment 16: 85 reported > presented

Response 16: Done (“reported” is changed with “presented”).

Comment 17: 94 shown (Table 1) that > shown (Table 1) that (Table 1 was already referred to in the previous sentence.)

Response 17: Done [sentence “Detailed characteristics like common name, country of origin and number of samples (without and with pesticide residues) of the analyzed samples are shown in Table 1.” is deleted]. This sentence is moved on the top of this section.

Comment 18: 126 samples (Table 2). > samples. (Table 2 was already referred to in the previous sentence.)

Response 18: Done [“(Table 2)“ from the second sentence is deleted].

Comment 19: 152-158 “Many samples… residues.” (These sentences should be moved below Table 3b.)

Response 19: Done (These sentences are moved below Table 3b).

Reviewer 2 Report

Comments and Suggestions for Authors

This article describes the results of pesticide residue testing of various fruits tested in Serbia by a modified QuEChERS LC MS/MS approach. A large percentage (63%) of the tested fruits had pesticide levels at or above 0.01 mg/kg, some of which were at or above the Serbian MRL (~5%). Grapefruits appear to show the highest levels of pesticides and the greatest number of total pesticides.

These results are interesting and useful, but the presentation is poor and there are significant missing information in the present article. 

Minor Points:

1. Line 6-7: Only the corresponding authors email is required, cleanup the affiliations

2. Line 36: Addition citations are required in addition to the WHO reference such as multiple review articles

3. Line 46: There only needs to be a single citation to the Serbian regulation

4. Line 50: Again only a single citation is needed for the EU regulation, additional citations including review articles would be appropriate

5. In this reviewers opinion, Table 1 and 2 could be moved to the supplementary - instead the authors should pull out the important data they wish to discuss.

6. As above, the Paragraph starting at Line 86 could be better summarized using Pie Charts in new Figure(s).

Major Points:

1. The methods are inadequately described. The authors cite a previous report on the method on rice and cabbage. The authors need to describe the method for each class of fruit. For example, was the method on whole fruit, flesh, peel, pulp, etc. This has severe implications for the results, especially considering that the fruits with the highest pesticide levels were citrus fruits where the peels are not consumed by humans.

2. There is a significant lack of references to existing reports on both the background in the introduction (that is lacking) as well as the method and the results and conclusions. The authors should rewrite the article to place these results in context with the existing body of research.

Author Response

Dear Reviewer,

Thank you very much for your comments and professional advice. These opinions help to improve our article. Based on your suggestion and request, we have made corrected modifications on the manuscript. Furthermore, we would like to show the details as follows:

Comment 1: This article describes the results of pesticide residue testing of various fruits tested in Serbia by a modified QuEChERS LC MS/MS approach. A large percentage (63%) of the tested fruits had pesticide levels at or above 0.01 mg/kg, some of which were at or above the Serbian MRL (~5%). Grapefruits appear to show the highest levels of pesticides and the greatest number of total pesticides.

Response 1: Yes, that is exactly what we concluded.

Comment 2: These results are interesting and useful, but the presentation is poor and there are significant missing information in the present article.

Response 2: We tried to make a significant improvement in the present article.

Minor Points:

Comment 3: 1. Line 6-7: Only the corresponding authors email is required, cleanup the affiliations.

Response 3: We made some changes in the affiliations part.

Comment 4: 2. Line 36: Addition citations are required in addition to the WHO reference such as multiple review articles.

Response 4: We inserted new references. New references are dealing with pesticides impacts on human health and the environment. In Supplementary Material 1 in Table S1 we presented pesticide residues in fruits that is obtained by other authors (in total 114 articles), including review articles. We have included a large number of papers from Supplementary material 1 in the article. It is evident that reviewers in the previous version of athe rticle did not receive Supplementary material 1 for review.

Comment 5: 3. Line 46: There only needs to be a single citation to the Serbian regulation.

Response 5: Pesticide regulations in Serbia were changed multiple times between 2016 and 2019. From that reason we mentioned all pesticide regulations from that period. We kept all the Serbian regulations.

Comment 6: 4. Line 50: Again only a single citation is needed for the EU regulation, additional citations including review articles would be appropriate.

Response 6: We have kept the EFSA annual reports covering the period from 2016 to 2019. In Supplementary Material 1 in Table S1 we presented pesticide residues in fruits that is obtained by other authors (in total 114 articles), including review articles. It is evident that reviewers in the previous version of article did not receive Supplementary material 1 for review.

Comment 7: 5. In this reviewers opinion, Table 1 and 2 could be moved to the supplementary - instead the authors should pull out the important data they wish to discuss.

Response 7: Both Tables are moved to the Results section, according to the request of reviewer 1.

Comment 8: 6. As above, the Paragraph starting at Line 86 could be better summarized using Pie Charts in new Figure(s).

Response 8: Sentences from Lines between 87 and 91 are presented in the Figures.

Major Points:

Comment 9: 1. The methods are inadequately described. The authors cite a previous report on the method on rice and cabbage. The authors need to describe the method for each class of fruit. For example, was the method on whole fruit, flesh, peel, pulp, etc. This has severe implications for the results, especially considering that the fruits with the highest pesticide levels were citrus fruits where the peels are not consumed by humans.

Response 9: The methods are described in detail. Only the following parts of the fruit such as stems, shell, caps and crown were not used for homogenization, according to EU and Serbian regulation. All of this is described in the article.

Comment 10: 2. There is a significant lack of references to existing reports on both the background in the introduction (that is lacking) as well as the method and the results and conclusions. The authors should rewrite the article to place these results in context with the existing body of research.

Response 10: We updated the Materials and Methods section, as well as the Results and Discussion. In the section ‘Results and Discussion’ we cited a large number of studies on the presence of pesticide residues in fruits.